# How Well do Feature Visualizations Support Causal Understanding of CNN Activations?

**Roland S. Zimmermann**[* 1]                     **Judy Borowski**[* 1]

**Robert Geirhos**[1]    **Matthias Bethge**[† 1]    **Thomas S. A. Wallis**[† 2]    **Wieland Brendel**[† 1]

[1] Tübingen AI Center, University of Tübingen, Germany.
[2] Institute of Psychology and Centre for Cognitive Science, Technical University of Darmstadt, Germany.
[*] Shared first authorship, determined by coin flip. `firstname.lastname@uni-tuebingen.de`
[†] Joint supervision.

## Abstract

A precise understanding of why units in an artificial network respond to certain stimuli would constitute a big step towards explainable artificial intelligence. One widely used approach towards this goal is to visualize unit responses via activation maximization. These synthetic feature visualizations are purported to provide humans with precise information about the image features that *cause* a unit to be activated — an advantage over other alternatives like strongly activating natural dataset samples. If humans indeed gain causal insight from visualizations, this should enable them to predict the effect of an intervention, such as how occluding a certain patch of the image (say, a dog's head) changes a unit's activation. Here, we test this hypothesis by asking humans to decide which of two square occlusions causes a larger change to a unit's activation. Both a large-scale crowdsourced experiment and measurements with experts show that on average the extremely activating feature visualizations by Olah et al. [40] indeed help humans on this task ($68 \pm 4\,\%$ accuracy; baseline performance without any visualizations is $60 \pm 3\,\%$). However, they do not provide any substantial advantage over other visualizations (such as e.g. dataset samples), which yield similar performance ($66 \pm 3\,\%$ to $67 \pm 3\,\%$ accuracy). Taken together, we propose an objective psychophysical task to quantify the benefit of unit-level interpretability methods for humans, and find no evidence that a widely-used feature visualization method provides humans with better "causal understanding" of unit activations than simple alternative visualizations.

## 1   Introduction

It is hard to trust a black-box algorithm, and it is hard to deploy an algorithm if one does not trust its output. Many of today's best-performing machine learning models, deep convolutional neural networks (CNNs), are also among the most mysterious ones with regards to their internal information processing. CNNs typically consist of dozens of layers with hundreds or thousands of units that distributively process and aggregate information until they reach their final decision at the topmost layer. Shedding light onto the inner workings of deep convolutional neural networks has been a long-standing quest that has so far produced more questions than answers.

One of the most popular tools for explaining the behavior of individual network units is to visualize unit responses via activation maximization [16, 33, 38, 35, 39, 36, 54, 15]. The idea is to start with an image (typically random noise) and iteratively change pixel values to maximize the activation

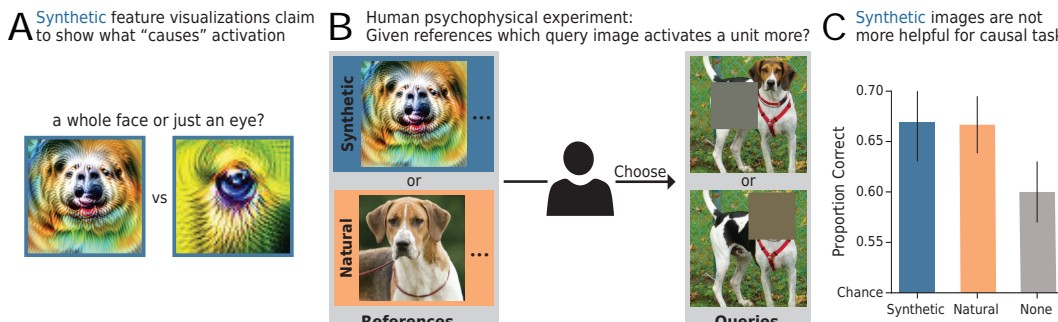

Figure 1: How useful are feature visualizations to interpret the effects of interventions? **A: "Causal" synthetic feature visualizations. B: Human experiment.** Given strongly activating reference images (e.g. synthetic or natural), a human participant chooses which out of two manipulated images activates a unit more. Note that this trial is made up — real trials are often more difficult. **C: Core result.** While participants are above chance for all visualization types, synthetic images only provide a substantial advantage over *no* references and not over other alternatives such as natural references.

of a particular network unit via gradient ascent. The resulting synthetic images, called *feature visualizations*, often show interpretable structures, and are believed to isolate and highlight exactly those features that "cause" a unit's response [40, 50]. Some of the synthetic feature visualizations appear quite intuitive and precise. As shown in Fig. 1A, they might facilitate distinguishing whether, for example, a unit responds to just an eye or a whole dog's face.

However, other aspects cast a more critical light on feature visualization's "causality": Generating these synthetic images typically involves regularization mechanisms [36, 33, 38, 35], which may influence how faithfully they visualize what "causes" a network unit's activation. Furthermore, to obtain a complete description of a mathematical function, one generally needs more information than just knowing its extrema. In view of this, it is an open question how well a unit can be characterized by simply visualizing the arguments of its maxima. Finally, a crucial unknown factor is whether *humans* are able to obtain a causal understanding of CNN activations from these synthetic visualizations.

Given these points, we develop a psychophysical experiment to test whether feature visualizations by Olah et al. [40] indeed allow humans to gain a causal understanding of a unit's behavior. Our task is based on the reasoning that being able to predict the effect of an intervention is at the heart of causal understanding. Understanding the causal relation between variables implies an understanding of how changes in one variable affect another one [45]. In our proposed experiment, this means that participants can predict the effect of an intervention — in form of an image manipulation — if they know the causal relation between image features and a unit's activations. Our experiment tests whether synthetic feature visualizations indeed provide information about such causal relations. Specifically, we ask humans which of two manipulated images activates a CNN unit more strongly. The interventions we test are obtained by placing an occlusion patch at two different locations in an image. Taken together, this experiment probes the purported explanation method's advantage of causality in a counterfactual-inspired prediction set-up [14].

Besides feature visualizations, other visualization methods have been used to gain an understanding of the inner workings of CNNs. In this experiment, we additionally test alternatives based on natural dataset examples and compare them with feature visualizations. This is particularly interesting because dataset examples are often assumed to provide less "causal" information about a unit's response as they might contain misleading correlations [40]. To continue the example above, dog eyes usually co-occur with dog faces; thus, separating the influence of one image feature from the other one using natural exemplars might be challenging.

Our data shows that:

- Synthetic feature visualizations provide humans with some helpful information about the most important patch in an image — but not much more information than no visualizations at all.
- Dataset samples as well as other combinations and types of visualizations are similarly helpful.
- How easily the most important patch is identifiable depends on the unit, the images as well as the relative activation strength attributed to the patch.

## 2 Related Work

**Feature visualizations** are a widely used method to understand the learned representations and decision-making mechanisms of CNNs [33, 38, 35, 39, 36, 54, 15, 40, 37]. As such, several works leverage this method to study InceptionV1 [42, 41, 8, 43, 50, 9, 58, 59, 46] and other networks [6, 21, 20]; others create interactive tools [61, 44, 52] or introduce analysis frameworks [65]. In contrast, some researchers question whether this synthetic visualization technique, first introduced by Erhan et al. [16], is too intuition-driven [27], and how representative the appealing visualizations in publications are [26]. Further, as already mentioned above, the engineering of the loss function may influence their faithfulness [36, 33, 38, 35]. Another challenge is generating *diverse* feature visualizations to represent the different aspects that one single unit may respond to [42, 36]. Finally, our recent human evaluation study [5] found that while these synthetic images do provide humans with helpful information in a forward simulation-inspired task, simple natural dataset examples are even more helpful.

**Human evaluation studies** are extensively used to quantify various aspects of interpretability. As an alternative to pure mathematical approximations [2, 66, 57, 63], researchers not only evaluate the understandability of explanation methods in psychophysical studies [7, 34, 5], but also trust in these methods [28, 64]) as well as the human cognitive load necessary for parsing explanations [1] or whether humans would follow an explained model decision [47, 13, 48]. A recent study even demonstrates that metrics of the explanation quality computed *with* human judgment are more insightful than those without [4].

**Counterfactuals** are a popular paradigm for both *creating* as well as *evaluating* explanation methods. Intuitively, they provide answers to the question "what should I change to achieve a different outcome?" — in the context of machine learning explanation methods, usually the smallest, realistic change to a data point is of interest. As examples, counterfactual explanation methods have been developed for vision- [22] and language-based [62] models as well as for model-agnostic scenarios [51]. Further, they are set into context of the EU General Data Protection Regulation [60]. Ustun et al. [56] investigate feasible and least-cost counterfactuals, while Mahajan et al. [32] and Karimi et al.

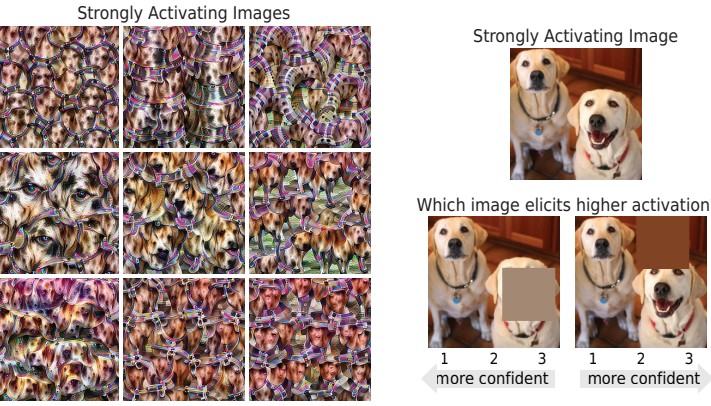

Figure 2: **Schematic visualization of an example trial** in our psychophysical experiment. For a certain network unit, participants are shown several maximally activating images. While the ones on the left serve as reference images, the ones on the right serve as query images: The top one is a natural maximally activating image and the bottom ones are copies of said image with square occlusions at different locations. The task is to select the image that activates the given network unit more strongly. Participants answer by clicking on the number below the corresponding image according to their confidence level (1: not confident, 2: somewhat confident, 3: very confident). Correct answer: right image.

[25] take feature interactions into account. To *evaluate* — rather than create — explanation methods, researchers often follow the "counterfactual simulation" task introduced by Doshi-Velez and Kim [14]: Humans are given an input, an output, and an explanation and are then asked "what must be changed to change the method's [model's] prediction to a desired output?" Doshi-Velez and Kim [14]. Based on this task, Lucic et al. [30] test their new explanation method and Hase and Bansal [24] compare different explanation methods to each other.

In this project, we design a counterfactual-inspired task to evaluate how well feature visualizations support causal understanding of CNN activations. This is the first study to apply such a paradigm to understanding the causes of individual units' activations. In order to scale the experiments, we

simplify our task by having participants choose between two intervention *options*, rather than having them freely determine interventions themselves.

## 3  Methods

We run an extensive psychophysical experiment with more than $12,000$ trials distributed over 323 crowdsourced participants on Amazon Mechanical Turk (MTurk) and two experts (the two first authors).[1] For more details than provided below, please see Appx. Sec. A.1.

**Design Principles**  Overall, our experimental design choices aim at (1) the *best performance possible*, meaning that we select images that make the signal as clear as possible; (2) *generality* over the network, meaning that we randomly sample units of different layers and branches (testing all units would be too costly); and (3) *easy extendability*, meaning that we choose a between-participant design (each participant sees only one reference image condition) so that other visualizations methods can be added to the comparisons in the future.

### 3.1  Psychophysical Task

If feature visualizations indeed support causal understanding of CNN activations, this should enable humans to predict the effect of an intervention, such as how occluding an image region changes a unit's activation. Based on this idea, we employ a two-alternative forced choice task (chance performance: $50\%$) where human observers are presented with two different occlusions in an image, and asked to estimate which of them causes a smaller change to the given unit's activation (see Fig. 2 for an example trial). More specifically, participants choose the *query* image that they believe to also elicit a strong activation given a set of 9 *reference* images. Such references could for instance consist of synthetic feature visualizations of a certain unit (purportedly "causal"), or alternative visualizations. To summarize, the task requires humans to first identify the shared aspect in the reference images and to then choose the query image in which that aspect is more visible. Since we do not make any assumptions about whether participants are familiar with machine learning, we avoid asking participants about activations of a unit in the CNN. Instead, we explain that an image would be "favored" by a machine, and the task is to select the image which is "more favored". The complete set of instructions shown to participants can be found in Appx. Fig. 9 and 10. In addition to each participant's image choice, the subjective confidence level and reaction time are also recorded.

### 3.2  Stimulus Generation

To generate stimuli, we follow Olah et al. [40] and use an InceptionV1 network [53] trained on ImageNet [12, 49]. Throughout this paper, we refer to a CNN's channel as a "unit" and imply taking the spatial average of all neurons in one channel.[2] We test units sampled from 9 layers and 2 Inception module branches (namely $3 \times 3$ and POOL). For more details on the generation procedures of the respective stimuli, see Appx. A.1.2.

We use five different types of **reference images**:

- **Synthetic references**: The synthetic images are the optimization results of the feature visualization method by Olah et al. [40] with the channel objective for 9 diverse images.

- **Natural references**: The reference images are the most strongly activating[3] dataset samples from ImageNet [12, 49].

- **Mixed references**: This is a combination of the previous two conditions: the 5 most strongly activating natural and 4 synthetic reference images are used. The motivation is that this condition combines the advantages of both worlds — namely precise information from feature visualizations and easily understandable natural images — and, thus, has the potential to give rise to higher performance in the task. Jointly looking at these two visualization types is common in practice [40].

---

[1] Code and data are available at github.com/brendel-group/causal-understanding-via-visualizations.

[2] Other papers might refer to a channel as a "feature map", e.g. [5].

[3] To reduce compute requirements, we use a random subset of the training set ($\approx 50\%$).

- **Blurred references**: To increase the informativeness of natural images for this task, we modify them by blurring everything but a single patch. This patch is chosen in the same way as in the maximally activating query image (see below). Consequently, this method cues participants to the most important image feature. In a way, these images can be seen as an approximate inverse of the maximally activating query image and might improve performance on our task.
- **No references**: This is a control condition in which participants do not see any reference images and have to solve the task purely based on query images.

To generate **query images**, we place a square patch of $90 \times 90$ pixels of the average RGB color of the occluded pixels into a most strongly activating image chosen from ImageNet. The location of the occlusion patch is chosen such that the activation of the manipulated image is either minimal or maximal among all possible occlusion locations. These images then yield the distractor and target query images respectively.

### 3.3  Structure of the Psychophysical Experiment

We test the five different reference image types as separate experimental conditions. In each condition, we collect data from a total of $50$ different MTurk participants, each assigned to a single Human Intelligence Task (HIT) consisting of an instruction block, a variable number of practice blocks and a main block. The instructions extensively explain a hand-crafted example trial (see Appx. Fig. 9 and 10). The blocks of $4$ practice trials each - which are randomly sampled from a pool of $10$ trials - have to be repeated until reaching $100\%$ performance; except in the none condition, as there is no obvious ground truth due to the absence of reference images. Finally, $18$ main trials follow that are randomly interleaved with a total of $3$ obvious catch trials. While feedback is provided during practice trials, no feedback is provided in the other trials. At the end, participants can share comments via an optional free-text field. Across all conditions, all participants see the same query images for the instruction, practice and catch trials. In contrast, the query images differ across participants in the main trials: In each reference image condition, we test $10$ different sets of query images, each responded to by $5$ different MTurk participants, hence $50$ HITs per condition. The order of the main and catch trials per participant is randomly arranged, and identical across conditions. Each MTurk participant takes part in only one reference image condition (i.e. reference images are a between-participants factor). For more details, see Appx. Sec. A.1.4.

### 3.4  Ensuring High-Quality Data in an Online Experiment

To ensure that the data we collect in our online experiment is of high quality, we take two measures: (1) We integrate hidden checks which were set before data collection. Only if a participant passes all five of them do we include his/her data in our analysis. First, these *exclusion criteria* comprise a performance threshold on the practice trials as well as a maximum number of blocks a participant may attempt. Further, they include a performance threshold for catch trials, a minimum image choice variability as well as a minimum time spent on both the instructions and the whole experiment. For more details, see Appx. Sec. A.1.1. (2) Our previous human evaluation study in a well-controlled lab environment found that natural reference images are more informative than synthetic feature visualizations when choosing which of two different images is more highly activating for a given unit [5]. We replicate this main finding on MTurk based on a subset of the originally tested units (see Appx. A.3) which indicates that the experiment's environment does not influence this task's outcome. Our decision to leverage a crowdsourcing platform is further corroborated by our result in Borowski et al. [5], that there is no significant difference between expert and lay performance.

### 3.5  Baselines

In order to both set MTurk participants' performance into context as well as evaluate different strategies participants could use to perform our task, we further evaluate a few baselines.

- **Expert Baseline**: The two first authors answer all $18$ trials in all $5$ reference conditions on all $10$ image sets. As they are familiar with the task design and are certainly engaged, this data serves as an upper human bound.
- **Center Baseline**: In natural images from ImageNet, important objects are likely to be closer to the center of the image. If participants were biased to assume that units respond to *objects*, a potential

strategy to decide which occluding patch produces a smaller effect on the unit's activation would therefore be to choose the image with the most eccentric occlusion. The Center Baseline model performs this strategy for all images.

- **Primary Object Baseline**: The Center Baseline is not a perfect measurement of an object-biased strategy because primary objects can appear away from the center. To account for this, the two first authors and the last author manually label all trials, choosing the image for which the occlusion hides as little information as possible from the most prominent object in the scene. In approximately one third of the trials ($58/180$), the authors' confidence ratings are very low (reflecting e.g. the absence of a primary object); in these cases we repeatedly replace the decisions by random binomial choices. Thus, in the results, we report the estimated expected values, but cannot perform a by-trial analysis. For more details, see Appx. Sec. A.1.3.

- **Variance Baseline**: Another assumption participants might make is that a patch in a low-contrast region, e.g. a blue sky, is unlikely to have a large effect on the unit's activation. This baseline selects the query image whose content is less affected by the introduction of the occlusion patch. To simulate this, we calculate the standard deviation over the occluded pixels and choose the one of the lower standard deviation.

- **Saliency Baseline**: As a complement to the baselines above, this baseline selects the query image whose original pixels hidden by the occlusion patch have a lower probability of being looked at by the participants. This simulates that participants select the image with a patch that occludes less prominent information and is estimated with the saliency prediction model DeepGaze IIE [29]. For more details, see Appx. Sec. A.1.3.

## 4    Results

The results shown in this section are based on $7350$ [4] trials from MTurk participants, who passed all exclusion criteria, and experts distributed over five conditions. In all figures, *Synthetic* refers to the purportedly "causal", activation-maximizing feature visualizations, *Natural* to ImageNet samples, *Mixed* to the combined presentation of synthetic and natural images, *Blur* to the blurred images, and *None* to the condition with no reference images at all. Further, error bars indicate two standard errors above and below the participant-mean over network units and image sets, unless stated otherwise.

### 4.1    No Significant Advantage of Synthetic Feature Visualizations

If feature visualizations provide humans with useful information about the image features causing high unit activations and other visualizations do not, participants' accuracy in our task should be higher given feature visualizations than for all other visualization types or no reference images. This is only partly what we find: On average, accuracy for feature visualizations is slightly higher than when no reference images are given ($67\pm4\%$ vs. $60\pm3\%$). However,

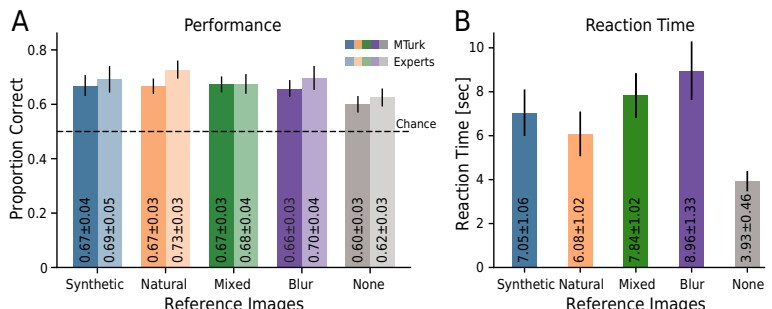

Figure 3: **A: Task accuracy.** On average, humans reach the same performance regime with any visualization method. This holds for both lay participants on MTurk (darker colors) as well as experts (brighter colors). **B: Reaction times.** MTurk participants need several seconds to answer a trial, indicating that they carefully make their decision. For more details see Appx. Fig. 13.

the accuracy for feature visualizations is not significantly higher than for other visualization methods (see Fig. 3A, dark bars). For the latter, MTurk participants reach between $66\pm3\,\%$ and $67\pm5\,\%$ depending on the visualization type. Statistically, only the condition without reference images is

---

[4]$(18$ main + 3 catch trials$)\times50$ MTurk participants $\times5$ conditions + $(18$ main + 3 catch trials$)\times20$ expert measurements $\times5$ conditions.

different from all other conditions ($p < 0.05$, Mann-Whitney U test). Taken together, these findings suggest that all visualization methods are similarly helpful for humans in our counterfactual-inspired task, and that they only seem to offer a small improvement over no visualizations at all.

### 4.1.1 MTurk Participants Carefully Make Their Choices

Similar performances for various conditions such as those found in Fig. 3A might suggest that participants would not give their best when doing our experiment. However, several aspects speak against this: (1) Measurement of the two first authors, i.e. experts who designed and thus clearly understand the task, and certainly engage during the experiment, again show very similar performance (see Fig. 3A, bright bars): This estimated upper bound is just $1 - 6\%$ better than MTurk participant performance. (2) With our strict exclusion criteria, we check for doubtful participant behavior and only include data from participants who pass all five criteria. (3) Reaction times per trial (see Fig. 3B) lie between $\approx 4\,\mathrm{s}$ and $\approx 9\,\mathrm{s}$. This, as well as the fact that participants take longer for the conditions *with* references than for the *None* condition, suggest that they carefully make their decisions. (4) Several MTurk participants' comments in an optional free-text field indicate that they engage in the task: "[...] I did my best", "It was engaging", "interesting task". (5) Trial-by-trial responses between MTurk participants are more similar than expected by chance (see Fig. 4B discussed below), which suggests that humans use the available information.

### 4.1.2 Simple Baselines Can Reach the Same Above-Chance Performance Regime

Decision-making strategies can be diverse. To set human performance into context, we evaluate several simple strategies as baselines: How high is performance if one always chooses the query image with an unoccluded center (Center Baseline) or primary object (Object Baseline)? Or such that the more varying or salient image region is unoccluded (Variance and Saliency Baseline)? Fig. 4A shows that these strategies have varying performances with the best ones — namely the Object and Variance baselines — reaching $63 \pm 1\%$ and $63\%$, respectively. Since already these simple heuristics, which do not require

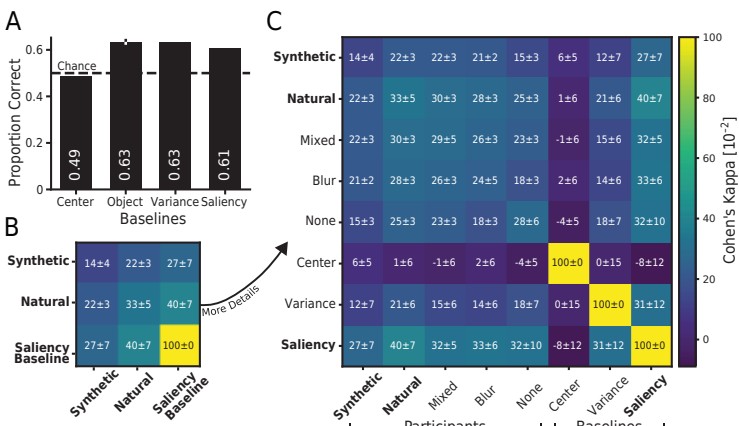

Figure 4: **A: Baseline performances.** Simple baselines can reach above chance level.[5] **B, C: Decision consistency.** The mean and two standard errors of the mean of Cohen's kappa averaged over participants and image sets quantifies the pairwise consistency of decision patterns.[6] While they vary across participants, they are higher between conditions with natural references and highest between the Saliency Baseline and other conditions. For more details, see Appx. Fig 15.

reference visualizations, can reach the same performance regime as participants, the additional advantage of visualizations (reaching just up to $4\%$ better performance) appears limited.

## 4.2 By-trial Decisions Show Systematic but Fairly Low Agreement

While accuracy is the most common metric to evaluate task performance, it does not suffice to compare two systems' decision-making processes [31, 19, 18]. Instead, a quantitative trial-by-trial error analysis is necessary to ascertain or distinguish strategies. Here, we use Cohen's kappa [10] to

---

[5]Only the Object Baseline has an error bar because in trials with, e.g. no clear primary object, we replace decisions by random binomial choices. The reported values are the estimated expectation value and standard deviation.

[6]There is no data for the Object Baseline because about one third of the trials do not have a clear answer from the three author responses. For more details, see Appx. A.1.3.

calculate the degree of agreement in classification while taking the expected chance agreement into account. A value of $1$ corresponds to perfect agreement, while a value of $0$ corresponds to as much agreement as would be expected by chance. Negative values indicate systematic disagreement.

In Fig. 4B and C, we plot consistency between MTurk participants of the same and different reference conditions as well as between MTurk participants and baselines. Since Cohen's kappa only allows for comparisons of two decision makers, we compute this statistic for all possible pairs across image sets, and report the mean over participants and image sets and two standard errors of the mean. All values between participants as well as between participants and baselines are in an intermediate regime (up to $0.40$). This suggests that there is systematic agreement, but also quite some room for subjective decisions. Among participant-baseline comparisons, highest agreement is found for the saliency baseline[7], while lowest agreement is found for the Center Baseline. Within participant to participant comparisons, decision strategies for conditions involving unmodified natural images (*Natural*, *Mixed*) are more similar to each other as well as slightly more similar to other strategies than the *Synthetic*, *Blur* or *None* condition to other strategies. Within the *Synthetic* condition, participants are relatively inconsistent. We hypothesize that due to the fact that humans are more familiar with natural images, they use more consistent information from these types of reference images and, thus, their decisions are more similar.

### 4.3 Performance Varies across Units, Image Sets and Activation Differences, but Less So for Reference Conditions

Having found that feature visualizations do not offer an overall advantage over other techniques, we now ask: Is performance similar across units, query images and their activation differences?

**Units and Image Sets**   As evident from Fig. 5, performance varies by unit, but usually not much by reference condition: While only one unit (layer 2, POOL) is clearly below chance level, many units reach around average performance and a few units stand out with high performances (e.g. layer 8, POOL). Further, the five reference conditions are relatively close to each other for most units. Finally, on the image set level, we observe fairly high variance - probably partly due to the limited number of participants per image set (see Appx. Fig. 14).

Fig. 6 further illustrates the different difficulty levels as well as the strong unit- and image-dependency: For the shown easy unit (Fig. 6A), the (presumably yellow-black) feature is fairly clearly identifiable and visible in the diverse reference and query images. In contrast, for the shown difficult unit (Fig. 6B), the unit's feature selectivity is unclear not only in the reference but also in the query images.

**Activation Differences**   We hypothesize that our task might be easier if the difference in activations between the two interventions of the query images is larger. In Fig. 7A and B, we plot by-image-set performance against the relative activation differences, i.e. the difference between activations elicited by the two manipulated images normalized by the unperturbed query image's activation. The figure shows that even though we select query images as the most strongly activating images for a unit, the relative activation differences vary widely. Furthermore, human performance indeed tends to increase with higher relative activation difference, confirming our hypothesis. This trend is stronger in the POOL than in the $3 \times 3$ branch as quantified by the Spearman's rank correlations in Fig. 7C.

## 5   Discussion & Conclusions

Explanation methods such as feature visualizations have been criticized as intuition-driven [27], and it is unclear whether they allow humans to gain a precise understanding of which image features "cause" high activation in a unit. Here, we propose an objective psychophysical task to quantify how well these synthetic images support causal understanding of CNN units. Through a time- and cost-intensive evaluation (based on $24,439$ trials taking more than $81$ participant hours including all pilot and reported experiments), we put this widespread intuition to a quantitative test. Our data provides no evidence that humans can predict the effect of an image intervention (occlusion) particularly well when supported with feature visualizations. Instead, human performance is only moderately above a

---

[7]From a different perspective, this result can be seen as a confirmation that the CNN learned to look at the "important" part of the image for downstream classification.

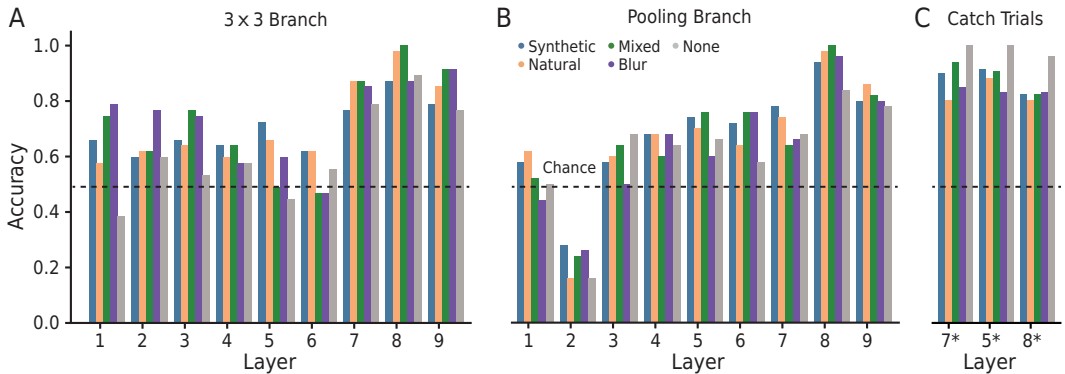

Figure 5: While for some units predicting the effect of an intervention is relatively easy, for most units performance is close to or just above chance. **A** and **B** show the **performance per unit** in the main trials separated by branch ($3 \times 3$ and POOL respectively) and layer. **C** shows the performance per unit in the hand-picked trials used as catch trials (hence the *), though selected from those MTurk participants who pass the exclusion criteria without the catch trial exclusion criterion. Note that each bar represents averages over participants and image sets.

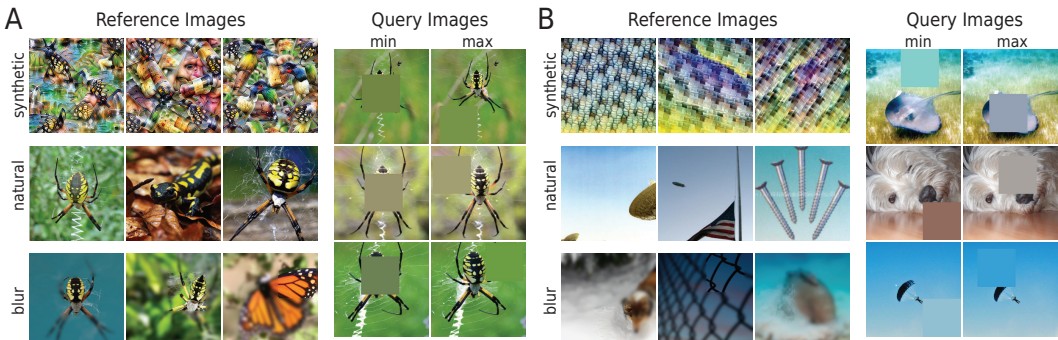

Figure 6: **Example reference and query images** for a unit with high (**A**) and low (**B**) performance from layer 8 and 2 of the POOL branch, respectively.

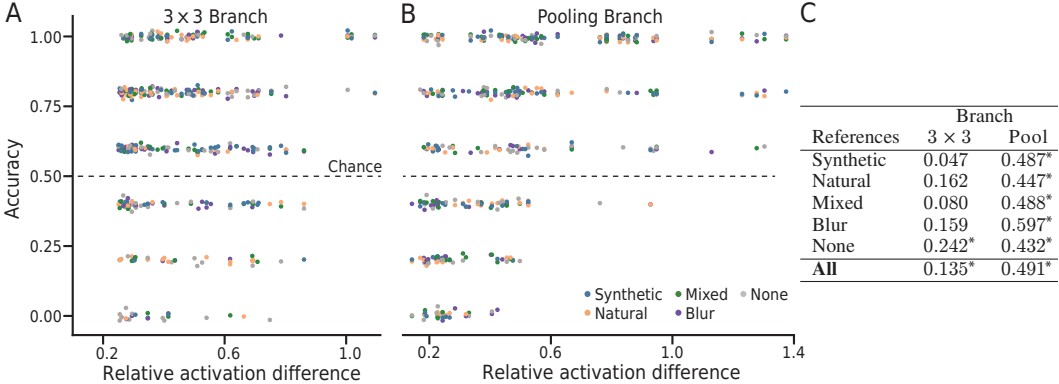

Figure 7: Performance tends to increase with the relative activation difference between query images. This effect is stronger for the POOL branch (**B**) than for the $3 \times 3$ branch (**A**) as quantified by Spearman's rank correlations (**C**). Stars signal significance ($p < .05$). Note that each dot in A and B represents the participant-averages, i.e. there is one dot per combination of layers, branch and image set. For an alternative visualization see Appx. Fig. 16.

baseline condition where humans are not shown any visualization at all, and similar to that of other visualization methods such as simple dataset samples. Further, by-trial decisions show systematic but fairly low agreement between participants. Finally, task performance depends on the unit choice, image selections and activation differences between query images. These results add quantitative evidence against the generally-assumed usefulness of feature visualizations for understanding the causes of CNN unit activations.

Our counterfactual-inspired task is the *first* quantitative evaluation of whether feature visualizations support causal understanding of unit activations, but it is certainly not the *only* possible way to evaluate causal understanding. For example, our interventions are constrained to occlusions of a fixed size and shape, imposing an upper limit on the precision with which the occlusions can cover the part of the image that is most responsible for driving a unit's activation. Future work could explore more complex intervention techniques, extend our study to more units of InceptionV1 as well as to different networks, and investigate additional visualization methods. Thanks to the between-participant design, new conditions can be added to the data without the requirement to re-run already collected trials.

Taken together, the empirical results of our quantitative evaluation method indicate that the widely used visualization method by Olah et al. [40] does not provide causal understanding of CNN activations beyond what can be obtained from much simpler baselines. This finding is contrary to wide-spread community intuition and reinforces the importance of testing falsifiable hypotheses in the field of interpretable artificial intelligence [27]. With increasing societal applications of machine learning, the importance of feature visualizations and interpretable machine learning methods is likely to continue to increase. Therefore, it is important to develop an understanding of what we can — and cannot — expect from explainability methods. We think that human benchmarks, like the one presented in this study, help to expose a precise notion of interpretability that is quantitatively measurable and comparable to competing methods or baselines. The paradigm we developed in this work can be easily adapted to account for other notions of causality and, more generally, interpretability as well. For the future, we hope that our task will serve as a challenging test case to steer further development of feature visualizations.

### Author Contributions

The idea to test how well feature visualizations support causal understanding of CNN activations was born out of several reviewer and audience comments on our previous paper [5]. The first idea of how to test this in a psychophysical experiment came from TSAW. JB led the project. JB, RSZ, WB and TSAW jointly improved the experimental set-up with input from MB and RG. RSZ led and JB helped with the implementation and execution of the experiment; JB led and RSZ contributed to the generation of stimuli. RSZ and JB both coded the baselines, and TSAW guided the replication experiment with statistical power simulations. The data analysis was performed by RSZ and JB with advice and feedback from RG, TSAW, WB and MB. TSAW and WB provided day-to-day supervision. While JB and RSZ created the first draft of the manuscript, RG and TSAW heavily edited the manuscript and all authors contributed to the final version.

### Acknowledgments

We thank Felix A. Wichmann and Isabel Valera for a helpful discussion. We further thank Ludwig Schubert for information on technical details via `slack.distill.pub`. In addition, we thank our colleagues for helpful discussions, and especially Matthias Kümmerer, Dylan Paiton, Wolfram Barfuss, and Matthias Tangemann for valuable feedback on our task, and/or technical support. Moreover, we thank our various reviewers and other researchers for comments on our previous paper inspiring us to investigate causal understanding of visualization methods. And finally, we thank all our participants for taking part in our experiments.

### Funding

The authors thank the International Max Planck Research School for Intelligent Systems (IMPRS-IS) for supporting JB, RSZ and RG. This work was supported by the German Federal Ministry of Education and Research (BMBF) through the Competence Center for Machine Learning (TUE.AI, FKZ 01IS18039A) and the Bernstein Computational Neuroscience Program Tübingen (FKZ 01GQ1002), the Cluster of Excellence Machine Learning: New Perspectives for Sciences (EXC2064/1), and the German Research Foundation (DFG, SFB 1233, Robust Vision: Inference Principles and Neural Mechanisms, TP3, project number 276693517). MB and WB acknowledge funding from the MICrONS program of the Intelligence Advanced Research Projects

Activity (IARPA) via Department of Interior/Interior Business Center (DoI/IBC) contract number D16PC00003. WB acknowledges financial support via the Emmy Noether Research Group on The Role of Strong Response Consistency for Robust and Explainable Machine Vision funded by the German Research Foundation (DFG) under grant no. BR 6382/1-1.

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
