# OpenReview forum: "How Well do Feature Visualizations Support Causal Understanding of CNN Activations?"
_NeurIPS.cc/2021/Conference — NeurIPS 2021 Spotlight_

### Official Review · Reviewer_yDAo · 2021-07-15

**Rating:** 8
**Confidence:** 4

**Summary:**

EDIT AFTER AUTHOR RESPONSE: I read the other reviews and I do not see the need to change my review.

The paper presents an experiment where users are trained, with or without the use of feature visualizations, to predict which of two images achieves higher activation on a unit from InceptionNet trained on ImageNet.  The query images are obtained by maximally activating and minimally activating occlusions of an natural image.  The general findings of the paper are that while feature visualizations help users get slightly higher accuracy than having no reference at all, there is no significant difference between synthetic visualizations and the highly activating natural references, and furthermore that user behavior is highly similar to the performance of a salience baseline that predicts user gaze on an image.


**Limitations And Societal Impact:**

Yes.

It is clear that this work is only a stepping stone, and that more studies are needed to investigate why human predictions of unit activations are so poor, and whether new tools can be developed to increase our ability to make predictions of neural network unit activations.


**Main Review:**

Originality:  Low. This work is similar to Borowski et al. 2021, but it does a more thorough evaluation encompassing more types of feature visualizations and more baselines.  It also uses an experimental design that is likely to be more powerful, as it compares two images that are obtained by occluding the same original image.  However it is not entirely clear that this methodological change is a strict improvement on Borowski et al. 2021 because such occluded distributions lie outside of the support of the natural image distribution and hence could fail to be representative of neural network activity for natural images, unless the network has been shown to be robust to occlusions.

Quality:  High. A lot of thought was put into the experimental design and the analysis of the conclusions.  In particular, decomposing the variation in accuracy by no only visualization method but also unit and contrast in activation helps build the case that the visualizations do not differ by much.  The details of collecting data on MTurk also build confidence in data quality.

Clarity:  The paper is clear and details of the methods are well-motivated and explained.

Significance:  Moderate.  The paper adds to the findings of Borowski et al. which pose serious questions for the feature visualization field.  Why are human rankings of feature activations so close to chance, and why don't visualizations (natural or synthetic) help more?


**Time Spent Reviewing:**

2

---

> ### Author Response · Authors · 2021-08-10
> **Response to Reviewer yDAo**
>
> Dear Reviewer,
>
> Thank you very much for your positive review. We are happy you found our paper to be high quality and thorough, and also “well-motivated and explained”.
>
> Please find our responses to your points below:
>
> **Comment**: _“Low originality”, “moderate significance” and “similar to Borowski et al. (2021)”_ \
> **Answer**: The study of Borowski et al. (2021) was criticized in its reviews and responses [1-3] for missing a crucial aspect of feature visualizations, namely that feature visualizations show the “causal link” between image feature and unit activation. This crucial aspect cannot be assessed with the task design and data from Borowski et al. (2021). Our paper aims to address this criticism through a “counterfactual-inspired” task. Therefore, it contributes to a more complete understanding of human evaluations of feature visualizations. As you pointed out, this might facilitate the development of new tools which “increase our ability to make predictions of neural network unit activations.”
>
> **Comment**: _Partially occluded images may lie outside of the “natural image distribution” and ”could fail to be representative of neural network activity for natural images”_ \
> **Answer**: We agree that this is an important consideration. While it is known that explanation methods may be o.o.d. (see feature visualizations vs. natural images in Appx. Fig. 10 of Borowski et al. (2021)), we accept the price of possibly o.o.d. query images for a counterfactual-inspired task design to address the causality aspect. \
> In response to your concern of whether this might limit the applicability of our results to network activity in natural images, we now evaluate how similar the response to the non-occluded and partially occluded images is (we count how often the top-5 predictions overlap). If network activations were drastically different for the occluded images, we would expect low accuracy. We found an accuracy of  ~97.8%. Therefore, the square occlusions only have a marginal effect on the network’s overall activity/predictions. We now include this “control experiment” in our paper, since other readers might have similar concerns.
>
> [1] https://openreview.net/forum?id=QO9-y8also- \
> [2] https://openreview.net/forum?id=-vhO2VPjbVa \
> [3] Olah, Chris. https://twitter.com/ch402/status/1321140564964765696 \
> [4] Engstrom, Logan, et al. "Adversarial robustness as a prior for learned representations." arXiv preprint arXiv:1906.00945 (2019).

---

> > ### Comment · Reviewer_yDAo · 2021-08-12
> > **Response**
> >
> > >  If network activations were drastically different for the occluded images, we would expect low accuracy. We found an accuracy of ~97.8%.
> >
> > Thanks for checking this, that is reassuring.

---

### Official Review · Reviewer_Nxjp · 2021-07-17

**Rating:** 6
**Confidence:** 4

**Summary:**

This paper put forward an objective psychophysical task to evaluate whether feature visualization could help humans understanding CNN activations. The authors recorded the details of their objective psychophysical task and adopted lots of baselines that ensure their experiments' preciseness. They also analyze the results from different angles.

**Limitations And Societal Impact:**

Besides the above mentioned, there are more concerns:

(1) More evidence should be provided to support that why this experiment is sufficient for supporting the claim. Why a similar performance between visualization and nature image can cause the uselessness of visualization explanation? Also, a unit may be sensitive to different patterns (multiclass classification), why nature image fixed in a single class?

(2) The authors do not illustrate how those reference images are assigned. Since the activation of CNN is relevant to units, the reference images should be different if units change. However, this paper doesn't show the details.

(3) It seems maximally activating synthetic image is full of one pattern, and it is understandable if the image is enlarged. However, when the synthetic image is the same size as the natural image, it is hard to recognize the pattern in the image. As the difference in query images is the location of the square patch, it is more reasonable to use a maximum activation patch as a reference rather than the same size image as the reference images.

(4) The authors point out that one advantage of this method is its easy extendability. However, If researchers want to investigate additional visualization methods,  they still need to generate new query images and re-run the collected trials.

(5) This paper lacks novelty, and the results are not interesting. Too many details which seem like placeholders because readers cannot get any inspiration from those paragraphs.



**Main Review:**

This is a brand-new idea that evaluates the performance of feature visualization totally with psychophysical experiments. However, it is more like a social science document rather than a research paper target for NeurIPS. This paper overflowed with design details, such as 3.1 Psychophysical Task, 3.3 Structure of the Psychophysical Experiment, and 3.4 Ensuring High-Quality Data in an Online Experiment, which could be paced in the supplement. What’s more, the results of the experiment also are trivial. Besides the task accuracy in experiment 1, all other experiments are hard to understand the purpose.

**Time Spent Reviewing:**

10 hours

---

> ### Author Response · Authors · 2021-08-10
> **Response to Reviewer Nxjp**
>
> Dear Reviewer,
>
> Thank you for reviewing our paper. While we are happy that you consider our idea “brand-new” and point out our experiments’ “preciseness”, you also mention a number of potential concerns/critical comments to which we respond below (ordered by topic):
>
> **Comment**: _“lacks novelty”_ \
> **Answer**: The claim that feature visualizations would show a “causal link” has been made several times in the literature (e.g. [1], [2]) as well as in public discussions [3]. However, this has, to the best of our knowledge, never been quantitatively assessed - and certainly not with human judges. As our study targets exactly this question, it is novel. If you are aware of previous work quantitatively assessing whether feature visualization provides causal insight to human observers, we would appreciate it if you could point us to specific references that substantiate your concern.
>
> **Comment**: _“results [...] are trivial” / “not interesting”_ \
> **Answer**: The method of feature visualization [1] has received widespread interest as evidenced by more than 500 citations since it was published in 2017. As indicated in the previous paragraph our work follows up on a public controversy about the claim that feature visualizations would show a “causal link” and is the first to quantitatively measure this. By doing so, it provides conclusive evidence against a causal advantage. Thus, we are confident that this study contributes to advancing the field of interpretability and it even fulfills the recent call for testing more falsifiable hypotheses for feature visualizations, as put forward by Leavitt and Morcos [4].
> If you disagree, could you please clarify? What previous work or logic would suggest that our results would be trivial?
>
> **Comment**: _“Purpose of experiments other than accuracy analysis is unclear"_ \
> **Answer**: Our various analyses rule out certain alternative hypotheses that could explain our main accuracy results. For example, it could have been possible that MTurk participants mostly ignore the reference images (and, as a result, would perform similarly for both natural and synthetic references). However, the reaction times clearly indicate that the participants did take the reference images into account.
>
> **Comment**: _“(2) The authors do not illustrate how those reference images are assigned. Since the activation of CNN is relevant to units, the reference images should be different if units change. However, this paper doesn't show the details.”_ \
> **Answer**: You find all information regarding the stimulus generation in Section 3.2 and Appx. A.1.2. To clarify this point, of course, we generate reference and query images for each unit individually. For  examples, please look at e.g. the butterfly and spider reference images in Fig. 6A third row for one unit, as well as the ray, dog and paraglider query images, and the reference images in row two and three of Fig. 6B for another unit. For even more illustrations, see Appx. Fig 13.
>
> **Comment**: _“too many details [...] 3.1 Psychophysical Task, 3.3 Structure of the Psychophysical Experiment, and 3.4 Ensuring High-Quality Data in an Online Experiment”_ \
> **Answer**: We intended to structure our paper clearly and write the text as succinctly as possible. All information to understand the general gist and fundamental design of our experiment is included in the main paper, details we considered necessary for reproducibility are placed in the Appendix. Given that Reviewer VeU5 states “The paper is extremely well-written and organized” and Reviewer yDAo also appreciated “The paper is clear and details of the methods are well-motivated and explained”, we decided to not change sections 3.1, 3.3 and 3.4. Nevertheless, we would be happy to consider specific suggestions on how we could improve the presentations if you and the other reviewers collectively agree.
>
> **Comment**: _This paper “is more like a social science document rather than a research paper target for NeurIPS”_ \
> **Answer**: While we agree that our project is of interdisciplinary nature, quantitative research at the intersection of machine learning and psychophysics does not make our paper a “social science document”. On the contrary, NeurIPS welcomes papers on “Social Aspects of Machine Learning” such as “interpretability” (see Call for Papers) and the Checklist specifically devotes an entire set of questions to research with human subjects, thus our paper clearly fits the scope of NeurIPS. Further, please consider the following non-exhaustive list of papers, demos and workshops at NeurIPS 2020, which collect data from humans, e.g. to evaluate explainability methods for machine learning models:
>
> - Jeyakumar, Jeya Vikranth, et al. "How can I explain this to you? An empirical study of deep neural network explanation methods."
> - Ramamurthy, Karthikeyan Natesan, et al. "Model agnostic multilevel explanations."
> - Plumb, Gregory, et al. "Regularizing black-box models for improved interpretability.
> - Demo at NeurIPS 2020: Shared Interest: Human Annotations vs. AI Saliency
> - Geirhos, Robert, et al. "Beyond accuracy: Quantifying trial-by-trial behaviour of CNNs and humans by measuring error consistency."
> - Zhang, Le, et al. "Disentangling Human Error from the Ground Truth in Segmentation of Medical Images."
> - Workshop: Shared Visual Representations in Human & Machine Intelligence
> - Crowd Science Workshop: Remoteness, Fairness, and Mechanisms as Challenges of Data Supply by Humans for Automation
> - Workshop: HAMLETS: Human And Model in the Loop Evaluation and Training Strategies
>
> **Comment**: _“(1) [...] Why [does] a similar performance between [synthetic] visualization[s] and natur[al] image[s] [suggest that feature visualizations do not provide humans with better ”causal understanding” of unit activations than simple alternative visualizations]?”_ \
> **Answer**: First, we assume that “if feature visualizations indeed support causal understanding of CNN activations, this should enable humans to predict the effect of an intervention, such as how occluding an image region changes a unit's activation” [line 245 ff.]. Then, from our collected data, we find that human performance is at 0.68 ± 0.04. As synthetic feature visualizations are purported to be more causal than natural dataset examples (since the latter might contain misleading correlations [1, 3]) one would expect higher human performance for synthetic feature visualizations than for natural dataset examples. However, this is not what our data shows: We find that performance is similar (namely 0.66 ± 0.05) and not significantly different from that of synthetic feature visualizations. Therefore, we conclude that “feature visualizations do not provide humans with better ‘causal understanding’ of unit activations than simple alternative visualizations” [line 19 f.].
>
> **Comment**: _“(1) [...] a unit may be sensitive to different patterns (multiclass classification), why nature[al] image fixed in a single class?”_ \
> **Answer**: There may be some misunderstanding here: natural images are not fixed in a single class -- in fact we completely ignore the class label for all images. The only selection criterion is the unit activation. And indeed, you are right that some units respond highly to different classes. For examples, see the butterfly and spider reference images in Fig. 6A third row for one unit , as well as the ray, dog and paraglider query images, and the reference images in row two and three of Fig. 6B for another unit. For even more illustrations, see the Appx. Fig 13.
>
> **Comment**: _“(3)” As feature visualizations repetitively show one aspect, enlarging that one aspect instead of showing the repetitions could be “more reasonable”_ \
> **Answer**: We understand the concern and can imagine this might come from the small images in the paper. In the real experiment, participants could only take part in the experiment if their screen resolution met a certain threshold which ensured that images would not be downscaled but are shown in high resolution. Therefore, repetitive patterns in the feature visualizations would have been clearly visible.
>
> **Comment**: _“(4)” extending the study requires generating new “query images and re-run[ning] the collected trials”_ \
> **Answer**: Thanks to our _between-participant_ study design, the effort to add new visualization methods is minimal: Only data for this new visualization method has to be collected; for other visualizations, our data can be used. This is in contrast to a _within-subject_ design, where each participant answers trials for all conditions, and adding a new visualization method would require indeed _re-running_ all visualizations methods. \
> Further, we plan to publicly release both our collected data as well as our source code after the anonymous review period and are happy to share all generated stimuli upon request to oblige with the license of ImageNet.
>
> [1] Olah, Chris, Alexander Mordvintsev, and Ludwig Schubert. "Feature visualization." Distill 2.11 (2017): e7. \
> [2] Schubert, Ludwig, et al. "High-Low Frequency Detectors." Distill 6.1 (2021): e00024-005. \
> [3] Olah, Chris. https://twitter.com/ch402/status/1321140564964765696 \
> [4] Leavitt, Matthew L., and Ari Morcos. "Towards falsifiable interpretability research." arXiv preprint arXiv:2010.12016 (2020).

---

> > ### Comment · Reviewer_Nxjp · 2021-08-30
> > **Thanks for the feedback and score raised**
> >
> > I would like to thank the authors for the efforts put into rebuttal. After reading the feedback and the comments from other reviews, I read the paper again.
> >
> > Though I still think the overall method is straightforward and lacks novelty, I agree that the authors put a lot of thought into experiments and it is helpful to better understand the causal explanation of feature visualization to humans.
> > As the feedback addressed some of my concerns. I am ready to increase my score.
> >
> > My remaining concerns:
> >
> > (1) I still have concerns about the reference images selection: as is shown in Fig. 1(B) and Fig.6 (A), when you using a natural image as a reference image, it provides not only one visual reference but also **category information** (e.g., dog in Fig. 1(B)), if the reference image is in the same class as the query images (e.g. Fig.1(B) and Fig.6 (A), both dogs or both spiders), participants may think the unit is used to identify a specific category (e.g. dog), and use their prior knowledge to choose the query image which keeps the most discriminative features to identify this specific class (e.g., keep the head of the dog because it is more discriminative to identify dog class and obtain larger activation value). In that situation, the results do not show their understanding of a particular unit, instead, the results show the probability that whether the most discriminative feature of a specific class aligned with your selected unit.
> >
> > On the contrary, the feature visualization of the maximum activation does not provide extra-label information, which causes unfairness.      I wonder how much proportion that the reference natural image has in the same class as the query image among all trails.
> >
> > (2) Though the authors use some implicit analysis (e.g., the time analysis) to claim that the participants do not give trivial solutions such as choosing only the center patches. It still lacks evidence to support that the participants can really understand the goal of the experiments especially if they can not understand "unit" and activation of CNN. Doing a binary choice is not that hard even though without correctly understanding the goal. If the results can not faithfully reflecting human's understanding of the feature visualization, then it is hard to value the contribution. Are there any more explicit or internal systems to guarantee the faithfulness or alarm the researcher if the participants make choices based on other rules?
> >
> > (3) It looks that the visualized reference images of synthesized feature visualization (e.g., Fig.2 and examples in Appendix) are too abstract to understand their meaning, which may hinder the participants to understand and really use it. The author uses limited kinds of feature visualization to support the conclusion that the whole feature visualization is not helpful to causal explanation. It may be too exaggerated.
> >
> > (4) If the author claims that the current feature visualization explanation can not help the understanding of causal explanation. Any more concrete suggestions to solve this problem or propose a better explanation method with better causal explanation ability?

---

> > > ### Author Response · Authors · 2021-09-01
> > > **Response 2 to Reviewer Nxjp**
> > >
> > > Dear Reviewer,
> > >
> > > Thank you for your response, and for raising your score. We will reply point by point in chronological order:
> > >
> > > **Comment**: *“(1) The experiment only tests if category-level information is provided in the images.” and “the feature visualization of the maximum activation does not provide extra-label information, which causes unfairness.”* \
> > > **Answer**: We agree that category-level information is probably among the most intuitive for humans. However, (I) our experiment does not test any category-specificity of units, and (II) an additional experiment shows that the overlap of classes between query and reference images is small. \
> > > (I) More specifically, our setup does not provide participants with any explicit category information but just shows images which are similar *as seen by the specific unit in question*. These units are *randomly sampled* across layers. As known from the literature (e.g. [1-3]), features in CNNs differ in their level of complexity across layers. Thus, by design, we have not imposed a category-bias. Instead, our setup examines whether participants can identify the most discriminative feature (as seen by the unit) and recognize it in the query image. We argue that this examines whether humans understand the responsiveness of units in a CNN. \
> > > As to Fig. 1B, we want to reiterate that this is a *fictional* trial created for illustrative purposes. We explain this in the caption and add that real trials are often more difficult. We created these straight-forward query images such that a reader would be able to quickly grasp the idea of the experiment. \
> > > As to Fig. 6A, we want to highlight that the shown unit fired most strongly for features present in spider, butterfly and bird images (not shown in the paper). It just so happens that many images of similar spider species contain the feature in question. In contrast, the unit shown in Fig. 6B clearly shows that images stem from a large variety of classes and that the feature in question does not contain any category-level information. For further examples, please see Appendix Fig. 13. Here, you can see that features seem to have to do with textures and other lower level features - which is exactly expected given the literature (e.g. [1- 3]). \
> > > (II) Finally, we ran an experiment to test your assumption that the reference and query images mostly come from the same classes: For each trial, we computed the number of reference images that have the same label as the query image and visualized these numbers as a histogram (see [here](https://i.imgur.com/NUboFwE.png)). Here, we see that for most trials the query and the large majority of the reference images come from different classes. We will add this plot to the appendix of the final manuscript.
> > >
> > > **Comment**: *“(2) lack evidence to support that the participants can really understand the goal of the experiments especially if they can not understand "unit" and activation of CNN”* \
> > > **Answer**: We are confident that the participants understood the task and were engaged in the experiment for a number of reasons: \
> > > (I) We include various quality checks as outlined in Section 3.4 (“Ensuring High-Quality Data in an Online Experiment”). Summarizing this section, we take three measures: First, we begin the experiment with a detailed and interactive explanation of the experiment. Here, the task is explained step by step to the participants (see Appx. Fig. 8 and Fig. 9). Second, before the participants can move on to the relevant trials, they have to achieve 100% accuracy on practice trials which we interpret as their successful understanding of the task. Third, we integrate checks throughout the experiment that exclude participants who did not work carefully on the tasks. \
> > > (II) Also, MTurk participants are **not** required to understand terms such as “unit” or “activation”. As we explain in Section 3.1., we break the task down and use appropriate vocabulary for our target participants. \
> > > (III) Besides the time analysis, we explain four other pieces of analyses in Section 4.1.1 which suggest that MTurk participants carefully make their choices (e.g. comparison to expert performance or differences between trials with and without reference images).
> > >
> > > **Comment**: *(3) Synthetic feature visualizations “are too abstract to understand their meaning”* \
> > > **Answer**: Thanks for pointing this out! We fully agree that some feature visualizations appear to be very abstract, which was part of our motivation for evaluating them in human trials. The feature visualizations tested in our experiments are precisely the ones claimed to yield causal insights into what units respond to, thus concerns regarding e.g. the abstractness of the tested visualizations apply to the visualization method itself, not to our analysis.
> > >
> > > **Comment**: *(3) ”The author uses limited kinds of feature visualization to support the conclusion that the whole feature visualization is not helpful to causal explanation. It may be too exaggerated.”* \
> > > **Answer**: The term “feature visualization” refers to methods that maximize unit activations to generate visualizations. There are a small number of different methods (most of which yield qualitatively similar results), but the one we use in our experiments is by far the most popular and widely used feature visualization method. Throughout our paper we make an effort to make clear statements and to not overgeneralize. As an example, the very final paragraph first summarizes our results with respect to the specific technique used, and only then do we speculate about the future role of the general method of feature visualizations.
> > >
> > > **Comment**: *(4) “Any more concrete suggestions to solve this problem [of a method granting causal insight] or propose a better explanation method with better causal explanation ability?”* \
> > > **Answer**: Great question! It is not obvious that purely maximizing causal insights in explainability methods is the best optimization objective. Rather, we believe explainability methods should be as informative as possible for humans. This may involve causal aspects but is definitely not constrained to them. In any case, future studies don’t need to speculate, hypothesize or conjecture whether certain novel methods lead to better explanations, since our objective psychophysical task will enable quantifying future improvements.
> > >
> > > [1] LeCun, Y., Bengio, Y., & Hinton, G. (2015). Deep learning. nature, 521(7553), 436-444. \
> > > [2] Güçlü, U., & van Gerven, M. A. (2015). Deep neural networks reveal a gradient in the complexity of neural representations across the ventral stream. Journal of Neuroscience, 35(27), 10005-10014. \
> > > [3] Goodfellow, I., Bengio, Y., & Courville, A. (2016). Deep learning. MIT press.

---

> > > > ### Comment · Reviewer_Nxjp · 2021-09-02
> > > > **Thanks for the feedback and updated review**
> > > >
> > > > I thank the authors for their detailed responses to my questions and comments. The response the authors posted helped me to understand both the motivation and details of the paper, especially the newly added experiments. I'm raising my rating from 4 to 6 as my concerns are well addressed in the response.

---

### Official Review · Reviewer_VeU5 · 2021-07-17

**Rating:** 7
**Confidence:** 4

**Summary:**

The paper designs and conducts a psychophysical experiment to evaluate how effective unit visualization is in interpreting a neural network. The experiments presents synthetic and natural images obtained through methods of unit activation maximization to human subjects and evaluate how these images could guide them in answering effects of occlusion of images to a network. Through extensive experiment controls at different levels, it arrives at the conclusion that, unit visualization with synthetic images is not particularly useful in helping us understand CNN activations.

**Limitations And Societal Impact:**

A potential limitation of the experiment is that, the occlusion based judgement design could bias human subjects to match objects or specific object parts to the occluded images. Units in a CNN, especially the ones in early layers, don't have to function as object detector or object descriptor detector. In this case unit visualization might still be helpful in identifying the general kind of feature that a unit cares about (for example, texture or the horizon) but fails to inform human subjects in the occlusion task.

It is not surprising to me that the saliency baseline yields similar performance compared to using maximally activated image and that the saliency baseline has the highest decision consistency with other conditions. This result can also be interpreted positively since it demonstrates that, through unit visualization, we verify that the network learns to look at the "important" part of the image for downstream classification. In another word, saliency models that are trained with human fixation data could be used not as a baseline but instead as a method in evaluating how well the network learns.


**Main Review:**

Originality: The psychophysical experiment design is novel and to my knowledge it is the first study that answers questions about unit interpretability in CNN with actual human behavioral data.

Quality: The amount of controls that are designed into the experiment is very impressive. It is able to rule out most of the competing hypothesis related to this problem. Analysis of the decision patterns, reaction time, as well as comparison to expert evaluator are also very informative.

Clarity: The paper is extremely well-written and organized.

Significance: The behavioral results in this paper has a lot of insights and is important in guiding the field for the next interpretability methods. It would also encourage more validating experiments to be done on other visualization methods in general.

**Time Spent Reviewing:**

3

---

> ### Author Response · Authors · 2021-08-10
> **Response to Reviewer VeU5**
>
> Dear Reviewer,
>
> Thank you very much for your positive review and your valuable feedback! It is very encouraging that you rate our work as “novel”, “very impressive” and “very informative”.
>
> Please find our responses to your points below:
>
> **Comment**: _“the occlusion based judgement design could bias human subjects to match objects or specific object parts to the occluded images”_.  \
> **Answer**: We understand your concern, but do not believe it critically influences our result. As you explained, early layers tend to focus on non-object-like features while later ones tend to function as object detectors. If the hypothesized bias was indeed influencing our participants’ performance, we should detect lower performance for early than for intermediate/later layers. As Fig. 5 A and B do not support this, we believe a potential bias does not play a large role. Additionally, as the query images in Appx. Fig. 13 A column 1, 5, 8 and 10 show, there _are_ images where square occlusions _can_ clearly highlight that a regular texture is the targeted (low-level) feature. \
> As our paper aims at testing the purported causal advantage of feature visualizations, occlusions (and their side-effects) are the price we pay to quantify this property in a psychophysical experiment. In addition, as pointed out in our discussion, square occlusions are perhaps not the ideal, but a practical and scalable choice. Sticking to purely natural query images would bring us back to a task as in Borowski et al. (2021). Their study showed that synthetic feature visualizations are helpful in a pure feedforward prediction task, but does not allow one to draw conclusions regarding the _causal_ advantage of this interpretability method over e.g. natural dataset examples, in terms of the local image features determining unit activation (as criticized by various reviewers / responses [1, 2, 3] to Borowski et al (2021)).
>
> **Comment**: _“the saliency baseline yields similar performance compared to using maximally activated images [... which] can also be interpreted positively”_ \
> **Answer**: Thank you for this interesting perspective on the saliency baseline. We will integrate it in the discussion of our results in the final version of the manuscript.
>
> [1] https://openreview.net/forum?id=QO9-y8also- \
> [2] https://openreview.net/forum?id=-vhO2VPjbVa \
> [3] Olah, Chris. https://twitter.com/ch402/status/1321140564964765696

---

> > ### Comment · Reviewer_VeU5 · 2021-08-17
> > **Response**
> >
> > Thank you for clarifying by pointing out results in Figure 5a and appendix. Despite good performance in earlier layers, it still seems like there are two "tiers" of performances for earlier and later layers. It will be interesting to see comparison of accuracy for images with occlusions on a regular texture versus objects.
> >
> > After reading other reviewers' response and author's responses to them, I would maintain my original assessment of the paper.

---

### Author Response · Authors · 2021-08-10
**Summary of Author Responses to Reviewer Comments**

We would like to thank all reviewers for their time and very much appreciate their assessment of our work as a _“well-motivated”_ (R3), _“well-written and organized”_ (R1) paper with _“high quality”_ (R3) and _“lot of insights”_ (R1). Our work was also acknowledged to contain a _“brand-new”_ experimental design (R2), an _“impressive”_ (R1) amount of experimental controls for building _“confidence in data quality”_ (R3) and a _“thorough”_ evaluation (R3) including _“lots of baselines”_ (R2). Finally our work was summarized to be _“important in guiding the field for the next interpretability methods”_ (R1). Several reviewer suggestions have been incorporated into the paper, which we believe further improved the manuscript.

Here is a summary of the main concerns and how we addressed them:
- high similarity between the saliency baseline and human responses (R1): We add this interesting perspective into our final manuscript.
- nature / interpretation of results (R2): We hope for a fruitful discussion with R2 as to how to further set our results into perspective with existing literature.
- partially occluded images may be o.o.d. (R3): In a follow-up experiment, we investigated how much the final predictions of partially occluded query images change compared to unmanipulated query images.

---

### Decision · Program_Chairs · 2021-09-27

**Decision:**

Accept (Spotlight)

**Comment:**

This paper investigates whether feature visualizations, such as activation maximization, provide a benefit to humans in predicting model performance over simpler approaches such as the maximally-activating data samples. By performing a set of human experiments, the authors show that a) there is a benefit to these visualizations over no visualizations, but that b) there is no difference between these approaches and the use of natural images. Reviewers all found the experiments to be very carefully designed and rigorous, and praised the overall clarity of the paper. There is certainly a need for more thorough and careful examination of the interpretability methods that are widely used, and I think this paper will prove impactful both because of its observations and as a model for how to conduct such evaluations. I recommend it be accepted as a spotlight.